# Positive Effects of Oral Antibiotic Administration in Murine Chronic Graft-Versus-Host Disease

**DOI:** 10.3390/ijms22073745

**Published:** 2021-04-03

**Authors:** Shinri Sato, Eisuke Shimizu, Jingliang He, Mamoru Ogawa, Kazuki Asai, Hiroyuki Yazu, Robert Rusch, Mio Yamane, Fan Yang, Shinji Fukuda, Yutaka Kawakami, Kazuo Tsubota, Yoko Ogawa

**Affiliations:** 1Department of Ophthalmology, Keio University School of Medicine, Tokyo 160-8582, Japan; shinri.sato259@keio.jp (S.S.); hejingliangai@126.com (J.H.); mamoogawa@gmail.com (M.O.); asai@keio.jp (K.A.); g.h.yazu@gmail.com (H.Y.); rusch.r.m@keio.jp (R.R.); enrich_mio@hotmail.com (M.Y.); yangfan_1991@sina.com (F.Y.); yoko@z7.keio.jp (Y.O.); 2Aier Eye school of Ophthalmology, Central South University, Changsha 410083, China; 3Department of Ophthalmology, Tsurumi University School of Dental Medicine, Kanagawa 230-0063, Japan; 4Institute for Advanced Biosciences, Keio University, Yamagata 997-0052, Japan; 5Transborder Medical Research Center, University of Tsukuba, Ibaraki 305-8575, Japan; 6Intestinal Microbiota Project, Kanagawa Institute of Industrial Science and Technology, Kanagawa 210-0821, Japan; 7Division of Cellular Signaling, Institute for Advanced Medical Research, Keio University School of Medicine, Tokyo 160-8582, Japan; yutakawa@keio.jp; 8International University of Health and Welfare School of Medicine, Chiba 286-0048, Japan

**Keywords:** graft-vs-host disease, antibiotics, microbiome, dry eye disease, gentamicin

## Abstract

Chronic graft-versus-host disease (cGVHD) is one of the most frequent complications experienced after allogeneic hematopoietic stem cell transplantation. Reportedly, dysbiosis and severe damage to the microbiome are also closely associated with GVHD. Herein, we aimed to elucidate the positive and negative effects of the administration of various antibiotics in a murine model of cGVHD. For allogeneic bone marrow transplantation (allo-BMT), bone marrow from B10.D2 mice were transplanted in BALB/c mice to induce cGVHD. The cGVHD mice were orally administered ampicillin, gentamicin (GM), fradiomycin, vancomycin, or the solvent vehicle (control group). Among the antibiotic-treated mice, the systemic cGVHD phenotypes and ocular cGVHD manifestations were suppressed significantly in GM-treated mice compared to that in control mice. Inflammatory cell infiltration and fibrosis in cGVHD-targeted organs were significantly attenuated in GM-treated mice. Although regulatory T cells were retained at greater levels in GM-treated mice, there were significantly fewer Th17 cells and interleukin (IL)-6-producing macrophages in cGVHD-targeted organs in these mice. Collectively, our results revealed that orally administered GM may exert positive effects in a cGVHD mouse model.

## 1. Introduction

Allogeneic hematopoietic stem cell transplantation (allo-HSCT) is an established treatment for hematologic malignancies and bone marrow failure disorders. Graft -versus-host disease (GVHD) is one of the major complications associated with allo-HSCT and can be categorized into acute (aGVHD) and chronic (cGVHD) types [1]. cGVHD arises from immune dysregulation several months after HSCT and is characterized by chronic tissue inflammation and fibrosis in cGVHD-susceptible organs; the changes noted include dry eye, lichen planus-like changes in the mouth, sclerotic features in the skin, stiffness in joints, bronchiolitis obliterans in the lung, strictures in the intestine, and cirrhosis in the liver [2,3]. Ocular GVHD can occur in approximately 50% of patients with cGVHD and primarily manifests as cGVHD-related dry eye disease [4]. Advanced ocular complications include corneal ulceration, corneal epithelial defects, secondary infection, perforation, stromal scarring, symblepharon, fornix shortening, and loss of vision [5,6,7]. As the number of long-term survivors who have undergone allo-HSCT increases, preventing deterioration in visual acuity is becoming critically important to control ocular GVHD [4,7].

Experimental studies have demonstrated the pathophysiological progression of aGVHD and cGVHD: (1) early inflammation caused by tissue injury and interaction between host antigen-presenting cells (APCs) and alloreactive cytotoxic T cells, (2) chronic inflammation, thymic injury, and dysregulated T-cell and B-cell immunity; and (3) culminating tissue repair with fibrosis [2,8]. Therefore, GVHD is primarily considered an immunological process of target organs. Moreover, as the link between the gut microbiota and systemic immune systems has been elucidated in studies, the impact of the gut microbiota on GVHD has received increasing attention. These studies, most of which have been conducted since the 1970s, have shown that the gut microbiota plays an important role in the pathophysiology of GVHD [9,10,11]. Recent studies in mice and humans have suggested important relationships between the microbiota and clinical outcomes in allo-HSCT recipients [12,13] and associations between specific bacteria and allo-HSCT outcomes have been observed repeatedly [14]. For example, patients with GVHD have more Proteobacteria and Firmicutes and fewer Bacteroidetes than those without GVHD [15,16]. The presence of *Enterobacteriaceae* and other Gram-negative bacteria from the phylum Proteobacteria has been linked to increased mortality, whereas a greater abundance of Lachnospiraceae, Actinomycetaceae, and the genus *Blautia*, which belong to the class Clostridia, has been linked to more favorable outcomes [17]. Short-chain fatty acids (SCFAs) produced by anti-inflammatory *Clostridia* (AIC) have been found to exhibit potency in inducing regulatory T cells (Tregs) and exert beneficial effects in patients undergoing allo-HSCT [13,18,19,20]. Patients undergoing allo-HSCT exhibited a predominance of lactic acid bacteria of the genus *Enterococcus* and enterococcal expansion in human patients, and the aGVHD mouse model was associated with the incidence of aGVHD and mortality [12]. The overall diversity of the gut microbiota reduces in patients undergoing allo-HSCT, and a high bacterial diversity is associated with improved overall survival after allo-HSCT [17,21]. 

The findings reported to date give rise to concerns on the practice of bacterial decontamination, such as via the administration of antibiotics, in patients undergoing allo-HSCT. Previous reports have indicated the positive and negative effects of antibiotic administration in patients undergoing allo-HSCT and in mouse models of allo-HSCT. Initial studies conducted in the 1970s in mouse models indicated that recipient germ-free mice that were treated using broad-spectrum antibiotics exhibited significantly reduced mortality [9,10,11]. In human studies, a prospective randomized trial revealed that the elimination of anaerobes by treatment with metronidazole and ciprofloxacin reduced GVHD [22]. A retrospective study conducted more recently has shown that gut decontamination before allo-HSCT increased the incidence of acute GVHD [23]. Currently, the precise effects of antibiotic administration on GVHD remain controversial and are under investigation. In addition, the previous reports focused on patients with aGVHD and murine models. Therefore, the effects of changes in the gut microbiota induced by oral antibiotic administration on cGVHD, particularly on the ocular cGVHD phenotype, are yet to be elucidated. 

In this study, we aimed to elucidate the positive and negative effects of various antibiotics on the aggravation of cGVHD, particularly in ocular examinations, using a cGVHD mouse model. We observed that gentamicin (GM) significantly attenuated the ocular manifestations of cGVHD and the expression of systemic cGVHD phenotypes, and also suppressed inflammatory cell infiltration and fibrosis in cGVHD-targeted organs. These findings indicate that GM could serve as a potential therapeutic agent in cGVHD, particularly for treating ocular complications associated with this condition.

## 2. Results

### 2.1. The Effects of Antibiotic Administration on the cGVHD Mice Model

#### 2.1.1. Systemic Clinical Phenotypes of Antibiotic-Treated cGVHD Mice

To examine the effects of oral antibiotic administration on cGVHD clinical phenotypes, we used an established sclerodermatous cGVHD mouse model [24,25]. These mice were treated per oral dose of either ampicillin (APBC), GM, fradiomycin (FRM), or vancomycin (VCM), and the control mice were administered antibiotic-free sterilized water. The treatment regime commenced 2 weeks before bone marrow transplantation (BMT) and continued until 4 weeks after BMT (Figure 1a). At 4 weeks after BMT, among the antibiotic-treated GVHD mice, the systemic GVHD scores of the GM-treated, ABPC-treated, and VCM-treated mice were significantly lower than the score of vehicle-treated mice (Figure 1b). Although no statistical significance was observed, the score of the systemic cGVHD phenotype of GM-treated mice was considerably lower than that of any other antibiotic-treated mice (Figure 1b). With respect to each sign of GVHD, weight loss, activity, fur texture, skin integrity, and presence of diarrhea were significantly attenuated in the GM-treated mice than in non-treated mice or mice (Appendix A). By 4 weeks post-BMT, the typical signs of the cGVHD phenotype, including blepharitis, skin keratinization, and diarrhea, observed in the non-treated cGVHD mice were more severe than that observed in GM-treated cGVHD mice (Figure 1b, right). 

#### 2.1.2. Ocular Clinical Phenotypes of Antibiotic-Treated cGVHD Mice

Four weeks after BMT, the corneal fluorescein score (CFS) values of GM-treated and ABPC-treated cGVHD mice were found to be significantly lower than that of non-treated control mice (Figure 1c.d, upper low). Next, we measured the tear-film breakup time (TFBUT) value, which is a key finding in dry eye disease and is clearly associated with a decline in visual performance and optical quality [26,27]. Four weeks after BMT, the GM-treated cGVHD mice had a significantly higher TFBUT value than that of the non-treated cGVHD mice (Figure 1d, middle low). Next, we measured tear secretion (TS), which is reduced drastically in patients with GVHD-related dry eye [26]. The reduction in TS, which was measured as the difference in TS measured 1 day before BMT and 4 weeks after BMT, in GM-treated and ABPC-treated cGVHD and syngeneic mice was significantly lesser than that in non-treated cGVHD mice (Figure 1d, lower low). Collectively, compared to non-treated cGVHD mice, GM-treated cGVHD mice showed significantly better results in the three ocular examinations, suggesting that oral GM administration may exert a positive effect on ocular cGVHD.

### 2.2. Attenuation of cGVHD by GM Treatment

#### 2.2.1. Pathological Analysis 

We examined the lacrimal glands (LGs), which is one of the most frequently affected organs in cGVHD [24,28]. Compared to the non-treated control mice, the GM-treated cGVHD mice showed fewer inflammatory cells in LGs, as observed using H&E staining (Figure 2a). In addition, we analyzed the fibrotic areas in the LGs and in the skin using Mallory staining. The fibrotic area per field in both LG and skin specimens was significantly suppressed in GM-treated mice compared to that in non-treated mice (Figure 2a–c). The fibrotic area per field in the LG specimens was significantly suppressed in GM-treated mice compared to FRM-, VCM-, and ABPC-treated mice and there was no significant difference between non-treated cGVHD mice and FRM-, VCM-, and ABPC-treated mice (Appendix A).

#### 2.2.2. GM-Induced Suppression of cGVHD-Related Molecules in cGVHD-Targeted Organs

To further examine inflammatory cell infiltration in cGVHD-targeted organs, we performed immunohistochemistry for the leukocyte marker cluster of differentiation (CD)45 [5]. More CD45^+^ inflammatory cells infiltrated the cGVHD-targeted organs (LGs, colon, liver, and lungs) in non-treated mice than in GM-treated cGVHD mice (Figure 3a). Among the antibiotics-treated mice, the number of CD45^+^ cells in the liver, lung and colon in GM-treated mice was significantly lower compared to other antibiotics-treated mice (Appendix A).

To further examine fibrosis in the cGVHD-targeted organs, we performed immunohistochemistry for evaluating the expression of the terminally differentiated fibroblast marker α-smooth muscle actin (α-SMA) [29,30] which is expressed at greater levels in systemic organs in the GVHD mice model for systemic sclerosis [5,31,32]. Consistent with the results of Mallory staining, the LGs, colon, liver, and lungs of GM-treated mice had fewer α-SMA^+^ cells per field that that in non-treated mice, suggesting that GM administration attenuated fibrosis in cGVHD-targeted organs (Figure 3b).

Th17 cells, which produce interleukin (IL)-17A, are associated with cGVHD, since the blockade of programmed death 1, which suppresses IL-17A-producing cells, attenuates the manifestations of cGVHD in the skin, liver, and salivary glands [33]. Increased IL-6 production during an autoimmune reaction leads to the induction of Th17 cells and a reduction in the Tregs population [34]. To examine the balance between Th17 cells and Tregs in GM-treated cGVHD mice, we analyzed the expression of Th17 and Treg markers. The number of IL-17^+^ CD4^+^ cells (Th17 cells) in LGs was significantly higher in the non-treated group than in the GM-treated group (Figure 4a), and the mRNA expression of *Il17a* in the LGs was significantly suppressed in GM-treated mice compared to that in non-treated control mice (Figure 4b), indicating that GM administration may suppress Th17 T cell differentiation in the LGs of cGVHD mice. The number of CD4^+^ CD25^+^ Foxp3^+^ cells (Tregs) in LGs was significantly greater in the GM-treated cGVHD mice than in the non-treated cGVHD mice (Figure 5a). Further, using flow cytometry analysis, we confirmed that a significantly higher percentage of CD4^+^ CD25^+^ Foxp3^+^ cells was present in the spleen in the GM-treated cGVHD mice than in the non-treated cGVHD mice (Figure 5b). These results indicate that GM administration might induce the production of Tregs and decrease the number of Th17 cells.

Next, we analyzed the levels of proinflammatory cytokines in GM-treated cGVHD mice. IL-6 is known to play a critical role in cGVHD progression [35,36] and an anti-mouse IL-6 receptor monoclonal antibody was shown to suppress cGVHD [5]. The mRNA expression of *Il6* was suppressed significantly in the LGs and lungs in GM-treated cGVHD mice compared to that in non-treated control mice (Figure 6a). We previously reported that the number of IL-6-producing activated macrophages in the LGs and spleen was significantly higher in cGVHD mice than in syngeneic control mice [5]. The number of IL-6^+^CD68^+^ macrophages in the cGVHD-targeted organs was higher in non-treated mice than in GM-treated cGVHD mice (Figure 6b). This suggests that GM administration may suppress the production of the proinflammatory cytokine IL-6 from activated macrophages in cGVHD mice.

## 3. Discussion

In this study, we demonstrated the positive effect of oral antibiotic administration in a mouse model of cGVHD. Among several antibiotics, GM appeared to most potent in terms of inhibiting cGVHD aggravation, as indicated by the significantly better scores for systemic cGVHD phenotypes and all three ocular examinations in GM-treated cGVHD mice than in non-treated cGVHD mice (Figure 1). In addition to the clinical phenotypes, in GM-treated mice, the cGVHD-targeted organs had fewer inflammatory cells and less intense fibrosis, as observed upon H&E and Mallory staining and immunohistochemistry for the leukocyte marker CD45 and the myofibroblast marker α-SMA (Figure 2 and Figure 3). These findings suggest that oral GM administration may have attenuated cGVHD. The local and/or systemic administration of glucocorticoids and/or immunosuppressants is the cGVHD treatment protocol currently adopted in clinical practice [3]. Local administration of glucocorticoids with lubricants is a major treatment method for ocular cGVHD; however, the overall response rate of this treatment method is limited to 43%–86% [37], and the side effects of glucocorticoid administration, such as glaucoma, cataract, and infection, can also pose a problem [7,38]. Therefore, the use of oral antibiotics may be a novel and effective approach for treating cGVHD, including its ocular manifestations, via regulation of the gut microbiome.

Recent studies have shown that commensal bacteria are critical for maintaining immune homeostasis in the intestine [39]. In this study, GM-treated cGVHD mice had fewer Th17 cells and more Tregs in their LGs than non-treated cGVHD mice did (Figure 4 and Figure 5). Several authors have reported bacterial species and metabolites that induce the differentiation of extrathymic T cells, including Tregs and Th17 cells [21,40]. Segmented filamentous bacterial antigens, presented by intestinal dendritic cells, have been shown to drive mucosal Th17 differentiation [40] and SCFAs, primarily butyric acid (and to a lesser degree, propionic and acetic acids), produced by AIC have been found to induce Treg production [13,18,19,20]. A recent human study has reported that the concentrations of the microbe-derived SCFAs in plasma samples from patients with developed cGVHD were lower compared with those without cGVHD manifestations [41]. The number of IL-6-producing macrophages in cGVHD-targeted organs was greater in non-treated cGVHD mice than that in GM-treated cGVHD mice (Figure 6). We previously reported that IL-6-producing macrophages that express the senescence marker Cdkn2a (p16) and cellular senescence-associated secretory phenotype (SASP) factors, including IL-6, promote ocular cGVHD [5]. Deoxycholic acid, a metabolite produced by Gram-positive gut bacteria, and lipoteichoic acid, a constituent of Gram-positive gut bacteria, promote hepatocellular carcinomas by upregulating the expression of SASP factors [42,43]. Therefore, we hypothesized that oral GM administration might alter the gut microbiota and affect T cell differentiation and/or cellular senescence. Oral GM administration in wild-type mice promotes the growth of *Bacteroides* and *Erysipelotrichaceae* [44]. Additionally, patients with GVHD have fewer Bacteroidetes than those without GVHD [15]. The growth of *Bacteroides* should be analyzed in GM-treated cGVHD mice microbiota. In future, 16S rRNA gene sequencing analysis of the intestinal microbiota should be performed to demonstrate how the gut microbiota is altered upon antibiotic administration, and the underlying molecular mechanism should also be investigated.

Although we did not analyze the systemic drug levels of GM, it is poorly absorbed from the gastrointestinal tract and is commonly used intravenously and intramuscularly [45,46]. Several authors have reported that the oral GM was used for gut decontamination in mice and humans [46,47,48] and no systemic side effects have been reported. Oral broad spectrum antibiotics administration (ampicillin + enrofloxacin and vancomycin + amikacin) causes the impaired post-BMT hematopoiesis due to the decrease in dietary energy uptake and the reduction in the visceral adipose tissue [49]. However, in our study, the body weight loss and diarrhea were significantly attenuated in GM-treated mice compared with non-treated mice, suggesting that impaired energy absorption might not occur in GM-treated cGVHD mice. Therefore, GM might have an effect mainly on the gut microbiome without a systemic activity and toxicity. Some antibiotics have the direct anti-inflammatory effect in addition to the antibacterial activity [50,51]. In the future, we plan to measure the systemic drug levels and take measures to rule out unknown effects of GM such as the fecal microbiota transplantation from wild-type donors with oral GM administration to cGVHD mice [52,53]. All data were obtained at the same time point, 4 weeks after BMT. It would be valuable to assess the magnification of cGVHD at multiple time points and long-term outcomes to know if antibiotic administration induced long lasting improvement or if the GVHD symptoms were simply delayed.

Recently, the effect of the gut microbiota on GVHD, particularly on aGVHD, has garnered attention, and studies on the intervention of gut microbial imbalances using antibiotic, prebiotic, probiotic, or postbiotic strategies are currently underway [14]. Here, we demonstrated the positive effect exerted by the oral antibiotic GM in the cGVHD mouse model, including the improvements in the ocular manifestations of cGVHD. Oral GM antibiotic administration may be a novel and clinically translatable strategy for attenuating the aggravation of cGVHD in susceptible organs, including the eyes. 

## 4. Materials and Methods

### 4.1. Mice

BALB/cCrSlc and B10.D2/nSnSlc mice (7–9 weeks old) were purchased from Sankyo Laboratory, Inc. (Tokyo, Japan). All animal experiments were in accordance with the ARVO Statement for the Use of Animals in Ophthalmic and Vision Research. The protocols for all experiments on animals were approved by the Keio University Institutional Animal Care and Use Committee (#09152).

### 4.2. Whole BMT 

To furnish the cGVHD mouse model, allo-BMT was performed using 7–9-week-old female BALB/cCrSlc and male B10.D2/nSnSlc mice as transplant recipients and donors, respectively, as previously reported [5,24]; the process represented MHC-compatible, minor histocompatibility (miHA)-mismatched BMT. To generate a non-GVHD control, BMT from male BALB/cCrSlc mice into female BALB/cCrSlc mice was conducted as syngeneic BMT. The recipients were irradiated with X-ray (7 Gy) using a Gammacell 137Cs source (Hitachi Medico, Ltd, Tokyo, Japan) prior to the BMT. Donor cells containing 2 × 10^6^ spleen cells/mouse and 1 × 10^6^ bone marrow cells/mouse were suspended separately in 100 μL of Roswell Park Memorial Institute (RPMI) 1640 medium and were then combined. The 200 μL suspension was administrated to each of the recipient mice via the tail vein [25]. The recipient mice were maintained in sterile cages and provided sterilized water and autoclaved. 

### 4.3. Treatment of Allo-BMT Recipient Mice with Antibiotics

The allo-BMT recipient mice were orally administered APBC (014-23302, Wako, Osaka, Japan), GM (073-06452, Wako), FRM (146-08871, Wako), or VCM (222-01303, Wako), and the control mice were orally administered the solvent vehicle. Sterilized drinking water was used as the medium. The treatments were administered starting 14 days before BMT and were continued until 28 days after BMT. The concentration of the antibiotic solutions was 0.5 mg/mL [54].

### 4.4. Assessment of GVHD

The mice were monitored for characteristic signs of GVHD by clinical assessment using a standard scoring system, as described previously [55,56]. This system assessed seven systemic clinical traits: activity, posture, fur texture, weight loss, skin integrity, presence of diarrhea, and degree of alopecia. Each clinical parameter was awarded a score from 0–2, with a total score of 0–14.

### 4.5. Histological Analysis 

The allo- and syngeneic-BMT recipient mice were sacrificed 4 weeks after the completion of BMT. The LGs, lungs, liver, colon, and skin of each mouse were harvested and fixed in 20% neutral buffered formalin for 2 h at 23 °C, embedded in paraffin wax, and processed for H&E, Mallory, and PAS staining [24,31]. Three images acquired from each Mallory-stained tissue sections of the LGs and skin were evaluated to quantify the fibrotic areas. We separated the colors of each image into blue, green, and red using Image J. The area with blue color was measured using the same threshold. The average value obtained for the three images from each sample was considered the value for the fibrotic area.

### 4.6. Immunostaining of Frozen Tissue Sections

The immunostaining of frozen tissue sections was performed as described previously [5,24]. The LG, lung, liver, and colon tissues were fixed in 10% neutral buffered formalin (Wako, Osaka, Japan) at 23 °C for 60 min, embedded in OCT compound (4583, Sakura Finetek, Torrance, CA, USA) in pre-cooled isopentane, and stored at −80 °C until the samples were cut into 6 μm-thick sections. The sections were blocked with 10% normal goat serum for 30 min at 23 °C and treated overnight at 4 °C with optimally diluted primary antibodies. In the experiment for α-SMA detection, a primary antibody specific for α-SMA (ab7817, 1A4, Abcam) was used. Next, the sections were treated for 45 min at room temperature with DAPI and Alexa Fluor 488-conjugated goat anti-mouse IgG secondary antibody (A11029, Thermo Fisher Scientific, Waltham, MA, USA). Each step was followed by three washes with PBS. For CD45 detection, the sections were stained with APC-labeled CD45 (103112, 30-F11, BioLegend, San Diego, CA, USA) and DAPI overnight at 4 °C. For detecting IL-6-producing macrophages, the sections were stained with APC-labeled CD68 (137007, FA-11, BioLegend) and PE-labeled IL-6 (554401, BD Biosciences, Franklin Lakes, NJ) antibodies and DAPI overnight at 4 °C. For Th17 cell detection, the sections were stained with APC-labeled IL-17A (17-7177-81, eBioscience, San Diego, CA) and FITC-labeled CD4 (17-0041-81, eBioscience) antibodies, along with DAPI, overnight at 4 °C. For Treg detection, the sections were treated with FITC-labeled CD4 (11-0041-81, eBioscience), PE-labeled CD25 (12-0251-82, eBioscience), and APC-labeled Foxp3 (103112, 30-F11, eBioscience) antibodies, along with DAPI, overnight at 4 °C. The number of target cells per field was quantified by averaging at least five non-overlapping fields for each section. 

### 4.7. RNA Isolation and Real-Time Quantitative Polymerase Chain Reaction (qPCR)

Total RNA was extracted as described previously [57] from the LGs using an miRNeasy mini kit (217004, Qiagen, Valencia, CA, USA). Complementary DNA was synthesized utilizing a ReverTra Ace qPCR RT Kit (FSQ-101, Toyobo, Osaka, Japan). Quantitative real-time PCR was performed using the StepOnePlus system (4379216, Thermo Fisher Scientific). The primers used for qPCR, including Mm00446190_m1 for *Il6* and Mm00439618_m1 for *ILl17a*, were purchased from Applied Biosystems/Thermo Fisher Scientific. The 2^−ΔΔCT^ method was used to analyze the data. To measure relative mRNA expression, the housekeeping gene *GAPDH* was used as the internal standard.

### 4.8. Flow Cytometry Analysis

The spleen of each cGVHD mouse was harvested at 4 weeks after BMT. The spleen cells were stained using a Treg Detection Kit (130-120-674, Miltenyi Biotic, Bergisch Gladbach, Germany). Single-cell suspensions of splenocytes were prepared by macerating the spleens with the rubber-coated end of a plunger from a 2.5 mL syringe and filtering the mixture through a 40 μM nylon filter (352340, BD Falcon, Bedford, MA, USA). Hemolysis was performed using ACK lysis buffer (5 mL/spleen) (118-156-721EA, Quality Biological, Inc., Gaithersburg, MD, USA). The cells were then washed with PBS and stained with APC-labeled anti-mouse CD25 antibody and Vio-Blue-labeled CD4 antibody for 10 min under refrigeration. The cells were then washed with PBS, permeabilized, and fixed using a fixation/permeabilization solution for 30 min in dark under refrigeration. After washing twice, the cells were treated with PE-labeled anti-mouse Foxp3 antibody for 30 min in dark under refrigeration. The stained cells were run on a flow cytometer (Gallios, Beckman Coulter Life Sciences, Indianapolis, IN, USA) and the data were analyzed using a software designed for the cytometer (Kaluza Analysis Software, Beckman Coulter Life Sciences).

### 4.9. Fluorescein Staining on the Ocular Surface

Four weeks after BMT, CFS was performed in order to assess the degree of corneal epithelitis. One microliter of 0.5% fluorescein sodium solution (Fluorescite, 877290, Novartis Pharma, Tokyo, Japan) was instilled into the temporal conjunctival sac. The ocular surface was observed under cobalt blue light using a Smart Eye Camera (13B2X10198030101, OUI Inc., Tokyo, Japan) [58]. The CFS value was calculated 90 s after fluorescein instillation. Each cornea was divided into four quadrants, and each quadrant was scored individually. The CFS value was scored based on a 0 to 4 grading scale: 0, absent; 1, slightly punctate staining with <30 spots; 2, punctate staining with >30 spots but not diffuse; 3, severe diffuse staining but no positive plaques; and 4, positive fluorescein plaques. The scores obtained from the four quadrants were summed to obtain a final grade (0–16 points) [59].

### 4.10. Cotton Thread Test for Measuring TS

Tears were measured by inserting a phenol red thread (Zone-Quick; 2564187, Showa Yakuhin Kako Co., Ltd., Tokyo, Japan) into the lateral canthus for 15 s. The length of the thread presenting with a change in color was measured for both eyes, and the average value was used to calculate the tear volume [57]. This procedure was performed on day 27 after BMT. 

### 4.11. Statistical Analysis

Data were analyzed using a two-tailed unpaired Student’s *t-*test. Differences are considered significant in the case of *p-*values <0.05. Data are presented as mean ± SEM. Statistical analyses were performed using Microsoft Excel 2013.

## Figures and Tables

**Figure 1 ijms-22-03745-f001:**
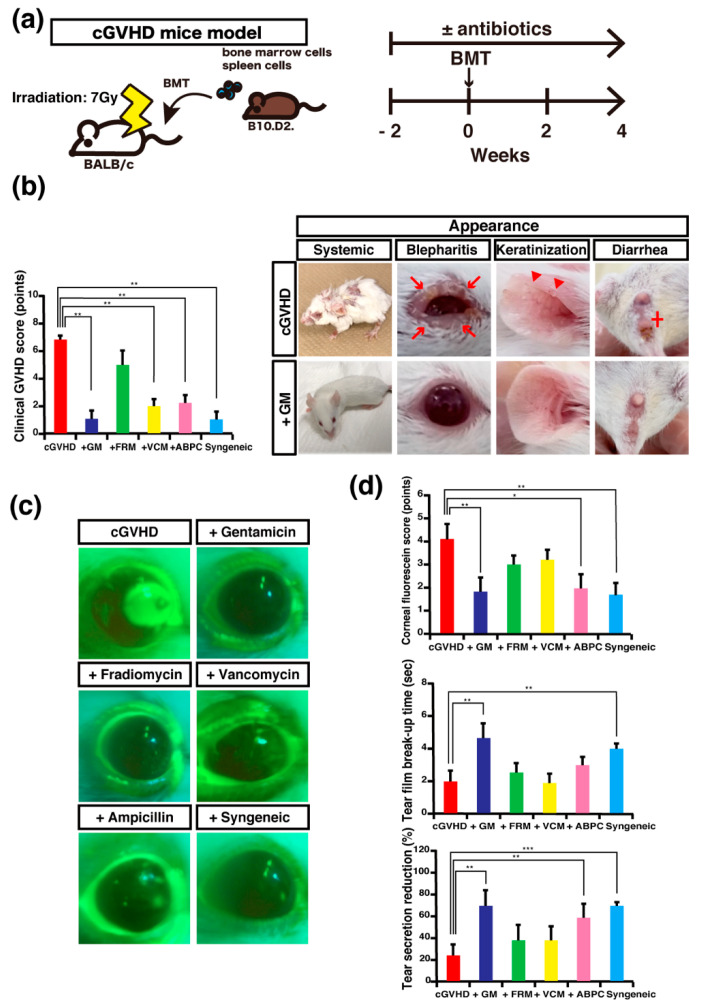
Systemic and ocular clinical phenotypes of antibiotic-treated chronic graft-versus-host disease (cGVHD) mice: (**a**) scheme of allogeneic bone marrow transplantation (BMT), and the experimental treatment protocol using antibiotics; (**b**) clinical GVHD scores of cGVHD mice treated with antibiotics, non-treated control mice, and syngeneic control mice (left) (*n* = 5 per group). GM, gentamicin; FRM, fradiomycin; VCM, vancomycin; ABPC, ampicillin. Data are presented as mean ± SEM. * *p* <0.05, unpaired Student’s *t*-test for non-treated cGVHD mice versus antibiotic-treated or syngeneic control mice. Representative images of non-treated and GM-treated cGVHD mice (+GM) (right). The non-treated cGVHD mice exhibited signs of cGVHD, such as blepharitis (red arrows), keratinization (red arrowheads), and diarrhea (red cross); (**c**,**d**) ocular examinations of cGVHD mice treated with antibiotics, non-treated control mice, and syngeneic control mice. Representative images of corneal fluorescein staining (**c**). Corneal fluorescein score (upper right), tear-film breakup time (middle right), and tear secretion reduction (lower right). Data are presented as mean ± SEM. * *p* <0.05, ** *p* <0.01, *** *p* <0.001, unpaired Student’s *t*-test for non-treated cGVHD mice versus antibiotic-treated or syngeneic control mice. (**a–d**), Data are representative of at least two independent experiments (*n* = 5 per group).

**Figure 2 ijms-22-03745-f002:**
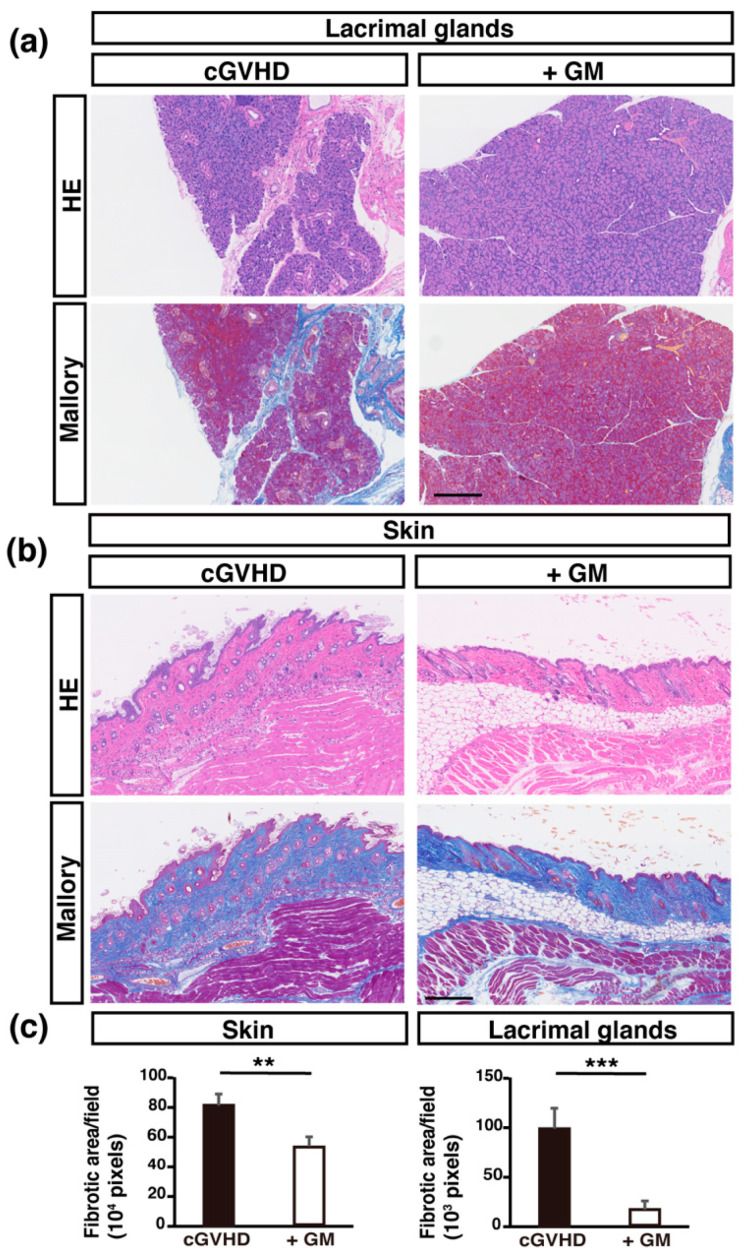
Pathological findings and analysis: (**a**,**b**) H&E and Mallory staining of lacrimal glands (LGs) and skin and from non-treated cGVHD mice (left) and gentamicin-treated cGVHD mice (+GM) (right); (**c**) Blue fibrotic areas indicated by Mallory staining were measured using ImageJ (*n* = 5 per group; 3 fields). Data are presented as mean ± SEM. ** *p* <0.01, *** *p* <0.001, unpaired Student’s *t*-test. Scale bar, 250 μm.

**Figure 3 ijms-22-03745-f003:**
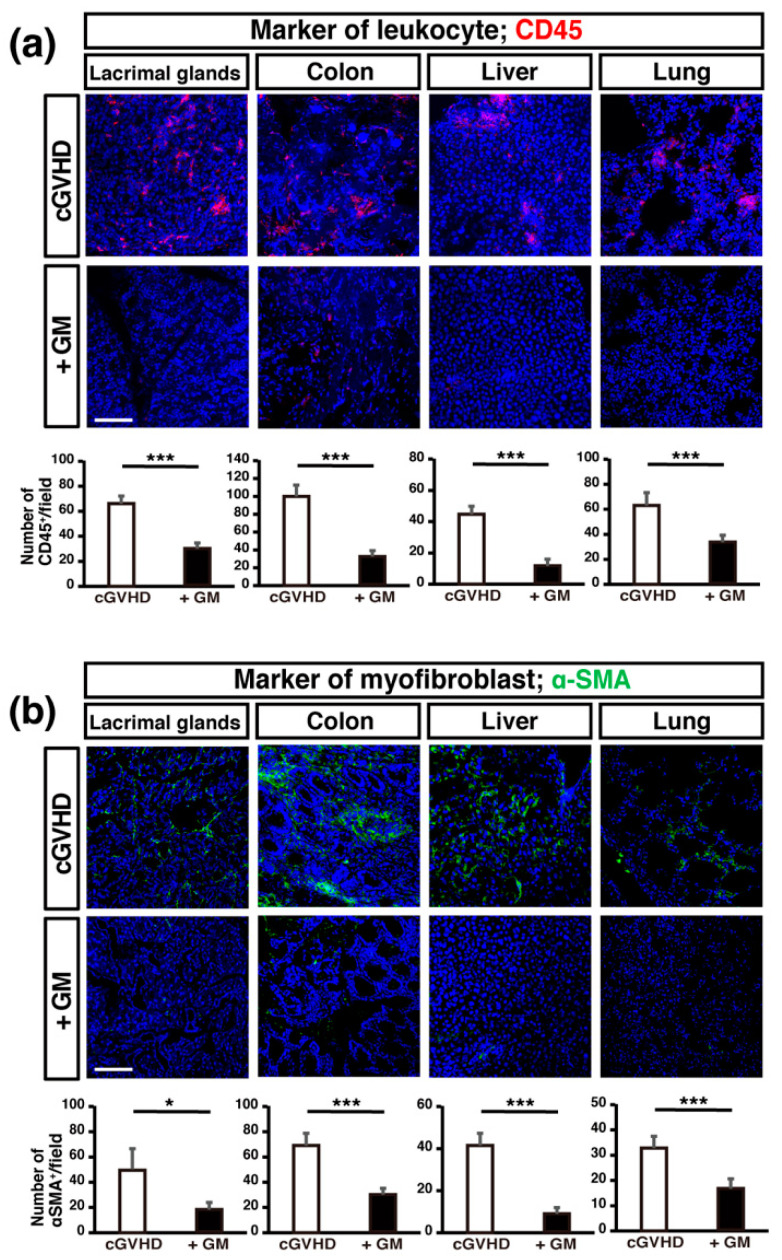
The expression of leukocyte and myofibroblast markers are suppressed in cGVHD-targeted organs in gentamicin-treated cGVHD mice: (**a**,**b**) sections of cGVHD-targeted organs (lacrimal glands, colons, livers, and lungs) from non-treated cGVHD mice (*n* = 5, five fields per sample) and gentamicin-treated cGVHD mice (+GM) (*n* = 5, five fields per sample) were stained for cluster of differentiation (CD)45 (leukocyte marker) and α-smooth muscle actin (α-SMA) (myofibroblast marker) (green). Data are presented as mean ± SEM. * *p* <0.05, *** *p* <0.001, unpaired Student’s *t*-test. Scale bar, 100 μm.

**Figure 4 ijms-22-03745-f004:**
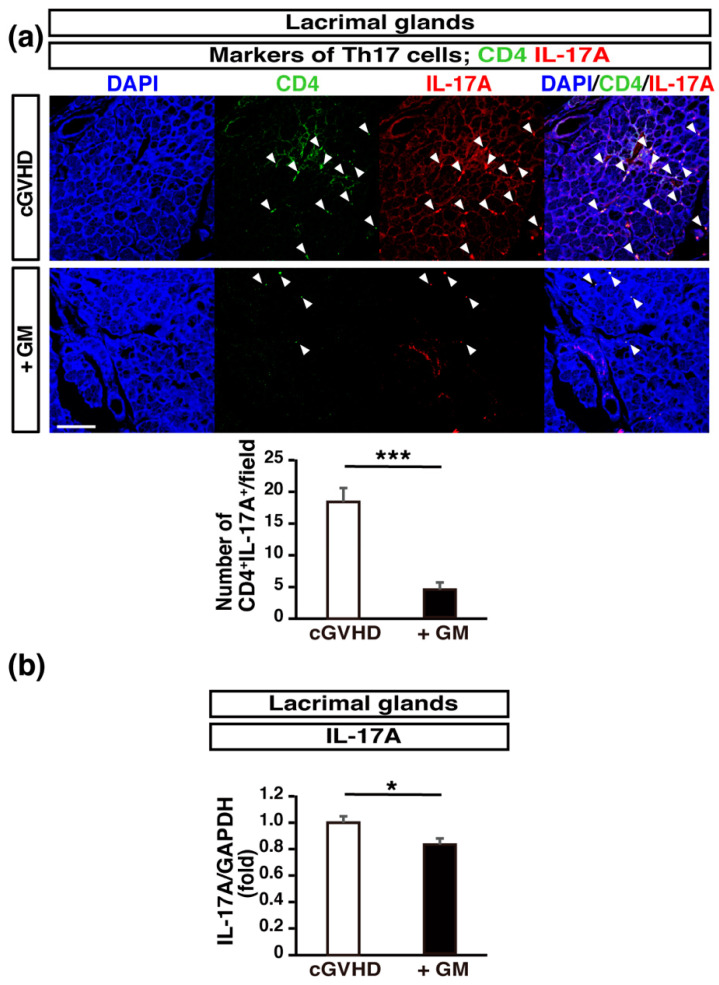
Th17 cell infiltration is suppressed in the lacrimal glands (LGs) of gentamicin-treated cGVHD mice: (**a**) sections of LGs from non-treated cGVHD mice (*n* = 5, five fields per sample) and gentamicin-treated cGVHD mice (+GM) (*n* = 5, five fields per sample) were stained for CD4 (green) and interleukin (IL)-17A (red). Data are presented as mean ± SEM. *** *p* <0.001, unpaired Student’s *t*-test. Scale bar, 100 μm; (**b**) *Il17a* expression in LGs. Quantitative real-time polymerase chain reaction analysis for *Il17a* in the LGs of non-treated (*n* = 5) and GM-treated (+GM) cGVHD mice (*n* = 5). Data are presented as mean ± SEM. * *p* <0.05, unpaired Student’s *t*-test.

**Figure 5 ijms-22-03745-f005:**
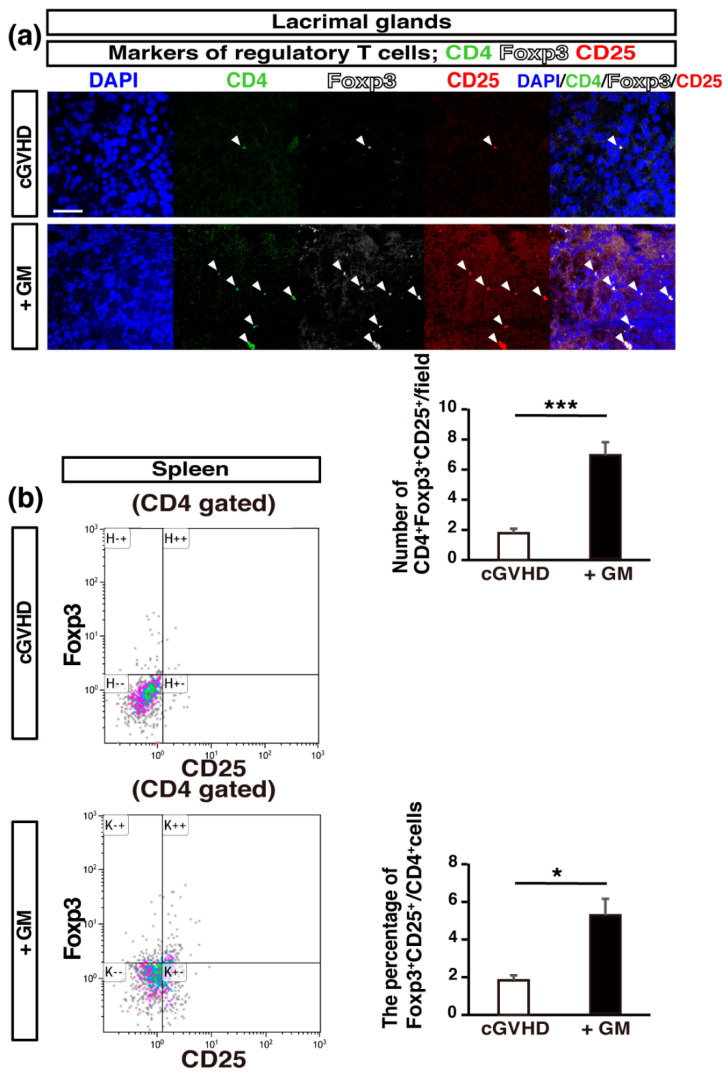
Regulatory T cells are retained in gentamicin-treated cGVHD mice: (**a**) sections of LGs from non-treated cGVHD mice (*n* = 5, five fields per sample) and gentamicin-treated cGVHD mice (+GM) (*n* = 5, five fields per sample) were stained for CD4 (green), Foxp3 (white), and CD25 (red). Data are presented as mean ± SEM. *** *p* <0.001, unpaired Student’s *t*-test. Scale bar, 50 μm; (**b**) flow cytometry analysis of regulatory T cells (CD4^+^ Foxp3^+^ CD25^+^ cells) in the spleen (*n* = 5 per group). Data are presented as mean ± SEM. * *p* <0.05, unpaired Student’s *t*-test.

**Figure 6 ijms-22-03745-f006:**
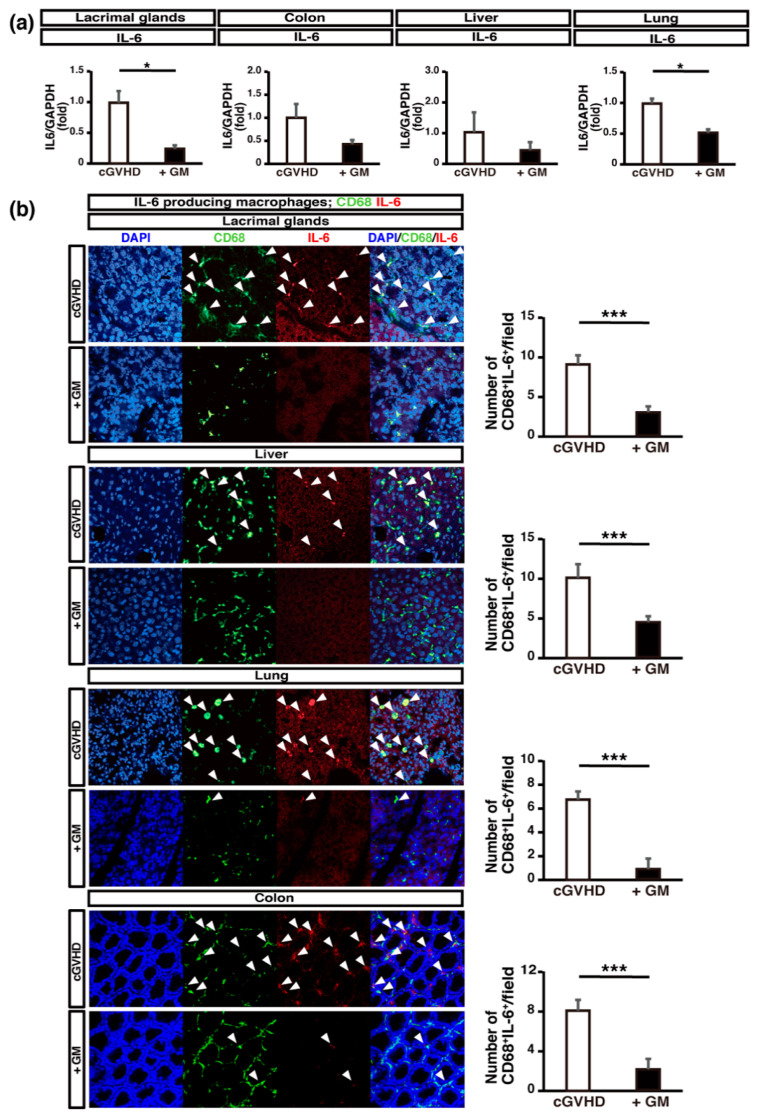
The infiltration of IL-6-producing macrophages in cGVHD-targeted organs is suppressed in gentamicin-treated mice: (**a**) *Il6* expression in cGVHD-targeted organs. Quantitative real-time polymerase chain reaction analysis of *Il6* in the organs of non-treated and gentamicin-treated (+GM) cGVHD mice. Data are presented as mean ± SEM. * *p* <0.05, unpaired Student’s *t*-test; (**b**) sections of LGs from non-treated cGVHD mice (*n* = 5, five fields per sample) and gentamicin-treated cGVHD mice (+GM) (*n* = 5, five fields per sample) were stained for CD68 (green) and IL-6 (red). Data are presented as mean ± SEM. *** *p* <0.001, unpaired Student’s *t*-test. Scale bar, 50 μm.

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
