# Peer review of "Positive Effects of Oral Antibiotic Administration in Murine Chronic Graft-Versus-Host Disease"

_ijms, 2021, doi:10.3390/ijms22073745_

Round 1

Reviewer 1 Report

The authors present an interesting manuscript on the effect of antibiotics on ocular cGVHD. The manuscript provides novel highly intersting results but some minor aspects need further improvement and the most relevant limitation is, that not microbiom analyses (16s sequencing) was performed and the results remain associative. Did the authors searched within the literature for effects of genta on the mouse microbiome outside the transplant setting?  This would indirectly support the data. Did the authors sampled faeces which would permit later analyses using 16s analysis? Do data exist on systemic drug levels of genta in oral application in mice? This would be important to exclude systemic effects outside the gut microbiome.

It would be helpful to provide in the results sections also the results of the other antibiotics since that supports that it is not simple antibiosis and the kind of antibiosis which makes a difference - some antibiotics could it even make worse. At least the authors should mention this in the text and if results are significantly different also in the figures. Just as example the authors show the results of genta on IL-6 but do not mention other antibiotic agents.

Minor:

Introduction first sentence: Please delete the word "protocol" - treatment for hematologic malignancies is sufficient.

Introduction: Would not use the term "Immune system activation" but "immune dysregulation"

Results section: The first sentence of the 2.1.2. part should be moved into the introduction or should be deleted.

The authors should discuss potential direct effects of antibiotics (some of them like doxycyclin have direct antiinflammatory effects which is unlikely the case for genta since it is hardly resorbed if given p.o. underlining the crucial role of the gut microbiome on inflammation in other organs.

Author Response

Dear reviwer,

Thank you for inviting us to submit a revised draft of our manuscript entitled, “Positive Effects of Oral Antibiotic Administration in Murine Chronic Graft-Versus-Host Disease” to IJMS. We also appreciate the time and effort you have dedicated to providing insightful feedback on ways to strengthen our paper. Thus, it is with great pleasure that we resubmit our article for further consideration. We have incorporated changes that reflect the detailed suggestions you have graciously provided. We also hope that our edits and the responses we provide below satisfactorily address all the issues and concerns you and the reviewers have noted.

To facilitate your review of our revisions, the following is a point-by-point response to the questions and comments delivered in your letter.

  1. Did the authors searched within the literature for effects of gentamicin on the mouse microbiome outside the transplant setting? This would indirectly support the data. 
  • RESPONSE: Thank you for your suggestion. We added a report which demonstrates that oral GM administration in wild-type mice promotes the growth of Bacteroides and Erysipelotrichaceae (lines 295-298 in Discussion section); Oral GM administration in wild-type mice promotes the growth of Bacteroides and Erysipelotrichaceae [1]. Also, patients with GVHD have fewer Bacteroidetes than those without GVHD [2] . The growth of Bacteroides should be analyzed in GM-treated cGVHD mice microbiota.

  1. Did the authors sampled faeces which would permit later analyses using 16s analysis?
  • RESPONSE: Thank you for your question. We sampled faeces from non-treated and antibiotics-treated cGVHD mice. As you suggested, 16s analysis is valuable. I have added the sentence about this in Discussion section (lines 298-301); In future, 16S rRNA gene sequencing analysis of the intestinal microbiota should be performed to demonstrate how the gut microbiota is altered upon antibiotic administration, and the underlying molecular mechanism should also be investigated.

  1. It would be helpful to provide in the results sections also the results of the other antibiotics since that supports that it is not simple antibiosis and the kind of antibiosis which makes a difference - some antibiotics could it even make worse. At least the authors should mention this in the text and if results are significantly different also in the figures.
  • RESPONSE: Thank you for your suggestion. We agree with you and the result from each antibiotics-treated cGVHD mice should be mentioned. I have added the results from pathological analysis in lacrimal glands (LGs) of each antibiotics-treated cGVHD mice (lines 168-171, Supplementary Materials Figure 1); The fibrotic area per field in the LG specimens was significantly suppressed in GM-treated mice compared to FRM-, VCM-, and ABPC-treated mice and there was no significant difference between non-treated cGVHD mice and FRM-, VCM-, and ABPC-treated mice.

Next, I have added the data showing the infiltration of CD45+ inflammatory cells in each antibiotics-treated cGVHD mice (lines 184-187, Supplementary Materials Figure 2); Among the antibiotics-treated mice, the number of CD45+ cells in the liver, lung and colon in GM-treated mice was significantly lower compared to other antibiotics-treated mice (Supplementary Materials Figure 2).

  1. Introduction first sentence: Please delete the word "protocol" - treatment for hematologic malignancies is sufficient.
  • RESPONSE: Thank you for your suggestion. I delated the word “protocol” (lines 42).
  1. Introduction: Would not use the term "Immune system activation" but "immune dysregulation"
  • RESPONSE: Thank you for your suggestion. I changed the word “Immune system activation” to “immune dysregulation” (lines 45).
  1. Results section: The first sentence of the 2.1.2. part should be moved into the introduction or should be deleted.
  • RESPONSE: Thank you for your suggestion. I delated the first sentence of 2.1.2.(lines 126).
  1. The authors should discuss potential direct effects of antibiotics (some of them like doxycyclin have direct antiinflammatory effects which is unlikely the case for genta since it is hardly resorbed if given p.o. underlining the crucial role of the gut microbiome on inflammation in other organs.
  • RESPONSE: Thank you for your suggestion. We have added the new sentence in Discussion section (lines 302-316); Although we did not analyze the systemic drug levels of GM, it is poorly absorbed from the gastrointestinal tract and is commonly used intravenously and intramuscularly [3, 4]. Several authors have reported the oral GM was used for the gut decontamination in mice and human [4-6] and no systemic side effects have been reported. Oral broad spectrum antibiotics administration (ampicillin + enrofloxacin and vancomycin + amikacin) causes the impaired post-BMT hematopoiesis due to the decrease of dietary energy uptake and the reduction of the visceral adipose tissue [7]. However, in our study, the body weight loss and diarrhea were significantly attenuated in GM-treated mice compared with non-treated mice, suggesting that impaired energy absorption might not occur in GM-treated cGVHD mice. Therefore, GM might have the effect mainly on gut microbiome without systemic activity and toxicity. Some antibiotics have the direct anti-inflammatory effect in addition to the antibacterial activity [8, 9]. In the future, we plan to measure the systemic drug levels and take measures to rule out direct or unknown effects of GM such as the fecal microbiota transplantation from wild-type donors with oral GM administration to cGVHD mice [10, 11] .

Again, thank you for giving us the opportunity to strengthen our manuscript with your valuable comments and queries. We have worked hard to incorporate your feedback and hope that these revisions persuade you to accept our submission.

  1. Zhao, Y.; Wu, J.; Li, J. V.; Zhou, N. Y.; Tang, H.; Wang, Y., Gut microbiota composition modifies fecal metabolic profiles in mice. J Proteome Res 2013, 12, (6), 2987-99.
  2. Noor, F.; Kaysen, A.; Wilmes, P.; Schneider, J. G., The Gut Microbiota and Hematopoietic Stem Cell Transplantation: Challenges and Potentials. J Innate Immun 2019, 11, (5), 405-415.
  3. van der Waaij, D.; Berghuis-de Vries, J. M.; Korthals Altes, C., Oral dose and faecal concentration of antibiotics during antibiotic decontamination in mice and in a patient. J Hyg (Lond) 1974, 73, (2), 197-203.
  4. Tascini, C.; Sbrana, F.; Flammini, S.; Tagliaferri, E.; Arena, F.; Leonildi, A.; Ciullo, I.; Amadori, F.; Di Paolo, A.; Ripoli, A.; Lewis, R.; Rossolini, G. M.; Menichetti, F., Oral gentamicin gut decontamination for prevention of KPC-producing Klebsiella pneumoniae infections: relevance of concomitant systemic antibiotic therapy. Antimicrob Agents Chemother 2014, 58, (4), 1972-6.
  5. Zuckerman, T.; Benyamini, N.; Sprecher, H.; Fineman, R.; Finkelstein, R.; Rowe, J. M.; Oren, I., SCT in patients with carbapenem resistant Klebsiella pneumoniae: a single center experience with oral gentamicin for the eradication of carrier state. Bone Marrow Transplant 2011, 46, (9), 1226-30.
  6. Brook, I.; Ledney, G. D., Oral aminoglycoside and ofloxacin therapy in the prevention of gram-negative sepsis after irradiation. J Infect Dis 1991, 164, (5), 917-21.
  7. Staffas, A.; Burgos da Silva, M.; Slingerland, A. E.; Lazrak, A.; Bare, C. J.; Holman, C. D.; Docampo, M. D.; Shono, Y.; Durham, B.; Pickard, A. J.; Cross, J. R.; Stein-Thoeringer, C.; Velardi, E.; Tsai, J. J.; Jahn, L.; Jay, H.; Lieberman, S.; Smith, O. M.; Pamer, E. G.; Peled, J. U.; Cohen, D. E.; Jenq, R. R.; van den Brink, M. R. M., Nutritional Support from the Intestinal Microbiota Improves Hematopoietic Reconstitution after Bone Marrow Transplantation in Mice. Cell Host Microbe 2018, 23, (4), 447-457.e4.
  8. Henehan, M.; Montuno, M.; De Benedetto, A., Doxycycline as an anti-inflammatory agent: updates in dermatology. J Eur Acad Dermatol Venereol 2017, 31, (11), 1800-1808.
  9. Buret, A. G., Immuno-modulation and anti-inflammatory benefits of antibiotics: the example of tilmicosin. Can J Vet Res 2010, 74, (1), 1-10.
  10. Bárcena, C.; Valdés-Mas, R.; Mayoral, P.; Garabaya, C.; Durand, S.; Rodríguez, F.; Fernández-García, M. T.; Salazar, N.; Nogacka, A. M.; Garatachea, N.; Bossut, N.; Aprahamian, F.; Lucia, A.; Kroemer, G.; Freije, J. M. P.; Quirós, P. M.; López-Otín, C., Healthspan and lifespan extension by fecal microbiota transplantation into progeroid mice. Nat Med 2019, 25, (8), 1234-1242.
  11. Biernat, M. M.; Urbaniak-Kujda, D.; Dybko, J.; Kapelko-Słowik, K.; Prajs, I.; Wróbel, T., Fecal microbiota transplantation in the treatment of intestinal steroid-resistant graft-versus-host disease: two case reports and a review of the literature. J Int Med Res 2020, 48, (6), 300060520925693.

Reviewer 2 Report

In the manuscript “Positive Effects of Oral Antibiotic Administration in Murine Chronic Graft-Versus-Host Disease”, Sato et al present an experimental study where oral administration of broad spectrum antibiotics improves cGVHD outcome after allo-BMT in mice. The study is generally clearly presented and focus on a clinically relevant question – how antibiotic administration affect development of chronic GvHD after allo-BMT. However, to clearly draw the stated conclusions I believe the experimental works needs to be more solid. I also have some concerns regarding the figures and statistics. Please see my comments below.

Major comments

  • The main conclusion of “positive effect of oral anitibiotic administration in a mouse model of cGVHD” would be more solid if based on more than one experiment. Figure 1 states n = 5 per group which I suspect is observations from one experiment? And while it could be argued that the groups administered other antibiotics could function as repeated observations, at least the effects of Gentamicin-treatment that is taken for further studies should be presented as a repeated observation.

  • The data in figure 1 (and the other?) are based on phenotype at one time point, 4 weeks after allo-BMT. It would be very valuable to know if the antibiotic treatment induced a chronic improvement or if the GVHD-symptoms were delayed?

  • In figure 1d, data is presented as mean±SEM. In all three panels the “cGVHD” and “+GM” have error bars that seem to overlap, yet the comparison is presented as P < 0.01 (**) by Students t-test. In general, SEM-bars that overlap indicate P > 0.05 (https://www.graphpad.com/support/faq/spanwhat-you-can-conclude-when-two-error-bars-overlap-or-dontspan/). Please correct if it is not SEM (but rather SD) that is shown or if the presented statistics are not correct. In general when the sample size is rather low, it is preferable to present individual data points together with mean±SEM to allow the reader to judge variation and the correctness of the statistical test performed (in this case a non-parametric t-test).

Minor comments

  • The study does not include any mechanistic evidence besides phenotypic observations of fewer immune cells in target organs and an increase in splenic Tregs. Since no mechanistic conclusions are made either, this is fine. But taken that reduced levels of immune cells are found in target organs the possible detrimental effects that oral broad spectrum antibiotics could have on donor reconstitution after BMT in mice (PMID: 29576480) could be discussed. Is expansion of donor cells inhibited (and thus GHVD-symptoms decreased/delayed) due to nutritional shortage upon antibiotic administration?

  • Another citation that should be made and which is supporting the message is that of a recent study on patients where decreased levels of short chain fatty acids correlate with cGVHD (PMID: 32430495)

  • Lastly, Figure 5b presents flow cytometry data of splenocytes. Percentage of Tregs (CD4+Foxp3+CD25+) are presented, but are the values (0.02 – 0.08) really percentages? They seem low?

Author Response

Dear reviewer,

Thank you for inviting us to submit a revised draft of our manuscript entitled, “Positive Effects of Oral Antibiotic Administration in Murine Chronic Graft-Versus-Host Disease” to IJMS. We also appreciate the time and effort you have dedicated to providing insightful feedback on ways to strengthen our paper. Thus, it is with great pleasure that we resubmit our article for further consideration. We have incorporated changes that reflect the detailed suggestions you have graciously provided. We also hope that our edits and the responses we provide below satisfactorily address all the issues and concerns you and the reviewers have noted.

To facilitate your review of our revisions, the following is a point-by-point response to the questions and comments delivered in your letter.

  1. The main conclusion of “positive effect of oral anitibiotic administration in a mouse model of cGVHD” would be more solid if based on more than one experiment. Figure 1 states n = 5 per group which I suspect is observations from one experiment? And while it could be argued that the groups administered other antibiotics could function as repeated observations, at least the effects of Gentamicin-treatment that is taken for further studies should be presented as a repeated observation.
  • RESPONSE: Thank your suggestion. The results of Figure 1 are representative of at least two independent experiments (n = 5 per group). We have added the referring information t (lines 156-157, figure legend of Figure 1); (a-d), Data are representative of at least two independent experiments (n = 5 per group).

I have attached the data below from the different experiment to show reproducibility. (a, b) The systemic clinical GVHD score and ocular examinations were significantly improved in GM-treated cGVHD mice compared to non-treated cGVHD mice. (c) The Inflammatory cell infiltration and fibrosis were attenuated in lacrimal glands of GM-treated mice.

  1. The data in figure 1 (and the other?) are based on phenotype at one time point, 4 weeks after allo-BMT. It would be very valuable to know if the antibiotic treatment induced a chronic improvement or if the GVHD-symptoms were delayed?
  • RESPONSE: As you mentioned, our data were based on phenotypes at a single time point, 4 weeks after BMT. We neither assessed long-term phenotypes nor those at multiple time points. Thus, this is limitation of our study and we have added the information in the discussion section (lines 316-320); All data were obtained at the same time point, 4 weeks after BMT. It would be valuable to assess the magnification of cGVHD at multiple time points and long-term outcomes to know if antibiotic administration induced long lasting improvement or if the GVHD symptoms were simply delayed.

  1. In figure 1d, data is presented as mean±SEM. In all three panels the “cGVHD” and “+GM” have error bars that seem to overlap, yet the comparison is presented as P < 0.01 (**) by Students t-test. In general, SEM-bars that overlap indicate P > 0.05 (https://www.graphpad.com/support/faq/spanwhat-you-can-conclude-when-two-error-bars-overlap-or-dontspan/). Please correct if it is not SEM (but rather SD) that is shown or if the presented statistics are not correct. In general when the sample size is rather low, it is preferable to present individual data points together with mean±SEM to allow the reader to judge variation and the correctness of the statistical test performed (in this case a non-parametric t-test).
  • RESPONSE: As you mentioned, the all error bars were SD. I have changed the SD to SEM in all figures not only in Figure 1d. Thank you for pointing out my mistakes.

  1. The study does not include any mechanistic evidence besides phenotypic observations of fewer immune cells in target organs and an increase in splenic Tregs. Since no mechanistic conclusions are made either, this is fine. But taken that reduced levels of immune cells are found in target organs the possible detrimental effects that oral broad spectrum antibiotics could have on donor reconstitution after BMT in mice (PMID: 29576480) could be discussed. Is expansion of donor cells inhibited (and thus GHVD-symptoms decreased/delayed) due to nutritional shortage upon antibiotic administration?
  • RESPONSE: Thank you for introducing us a valuable report (PMID: 29576480). We have added it to the discussion section (lines 306-311): Oral broad spectrum antibiotics administration (ampicillin + enrofloxacin and vancomycin + amikacin) causes the impaired post-BMT hematopoiesis due to the decrease of dietary energy uptake and the reduction of the visceral adipose tissue [1]. However, in our study, the body weight loss and diarrhea were significantly attenuated in GM-treated mice compared with non-treated mice in GM-treated mice, suggesting that impaired the energy absorption might not occur in GM-treated cGVHD mice.

  1. Another citation that should be made and which is supporting the message is that of a recent study on patients where decreased levels of short chain fatty acids correlate with cGVHD (PMID: 32430495)
  • RESPONSE: Thank you for suggesting us another report. We have added this report as well (lines 284-286): A recent human study has reported that the concentrations of the microbe-derived SCFAs in plasma samples from patients with developed cGVHD were lower compared with those without cGVHD manifestations [2].
  1. Lastly, Figure 5b presents flow cytometry data of splenocytes. Percentage of Tregs (CD4+Foxp3+CD25+) are presented, but are the values (0.02 – 0.08) really percentages? They seem low?
  • RESPONSE: Thank you for your question. The values (0.02-0.08) were the percentages of Tregs (CD4+Foxp3+CD25+) among total spleen cells. We have changed the data to the percentage of Foxp3+CD25+ among CD4+ cells (1.93 % in non-treated cGVHD mice and 5.25% in GM-treated cGVHD mice) (Figure 5b). Compared with the previous report (the percentage of CD25+Foxp3+ cell among CD4+ cells in cGVHD mice was around 6-7%) [3], those percentages were low. However, the number of CD4+Foxp3+CD25+ cells was around 100 per single spleen in GM-treated mice and this is enough to analyze.

Again, thank you for giving us the opportunity to strengthen our manuscript with your valuable comments and queries. We have worked hard to incorporate your feedback and hope that these revisions persuade you to accept our submission.

  1. Staffas, A.; Burgos da Silva, M.; Slingerland, A. E.; Lazrak, A.; Bare, C. J.; Holman, C. D.; Docampo, M. D.; Shono, Y.; Durham, B.; Pickard, A. J.; Cross, J. R.; Stein-Thoeringer, C.; Velardi, E.; Tsai, J. J.; Jahn, L.; Jay, H.; Lieberman, S.; Smith, O. M.; Pamer, E. G.; Peled, J. U.; Cohen, D. E.; Jenq, R. R.; van den Brink, M. R. M., Nutritional Support from the Intestinal Microbiota Improves Hematopoietic Reconstitution after Bone Marrow Transplantation in Mice. Cell Host Microbe 2018, 23, (4), 447-457.e4.
  2. Markey, K. A.; Schluter, J.; Gomes, A. L. C.; Littmann, E. R.; Pickard, A. J.; Taylor, B. P.; Giardina, P. A.; Weber, D.; Dai, A.; Docampo, M. D.; Armijo, G. K.; Slingerland, A. E.; Slingerland, J. B.; Nichols, K. B.; Brereton, D. G.; Clurman, A. G.; Ramos, R. J.; Rao, A.; Bush, A.; Bohannon, L.; Covington, M.; Lew, M. V.; Rizzieri, D. A.; Chao, N.; Maloy, M.; Cho, C.; Politikos, I.; Giralt, S.; Taur, Y.; Pamer, E. G.; Holler, E.; Perales, M. A.; Ponce, D. M.; Devlin, S. M.; Xavier, J.; Sung, A. D.; Peled, J. U.; Cross, J. R.; van den Brink, M. R. M., The microbe-derived short-chain fatty acids butyrate and propionate are associated with protection from chronic GVHD. Blood 2020, 136, (1), 130-136.
  3. Ogawa, Y.; Morikawa, S.; Okano, H.; Mabuchi, Y.; Suzuki, S.; Yaguchi, T.; Sato, Y.; Mukai, S.; Yaguchi, S.; Inaba, T.; Okamoto, S.; Kawakami, Y.; Tsubota, K.; Matsuzaki, Y.; Shimmura, S., MHC-compatible bone marrow stromal/stem cells trigger fibrosis by activating host T cells in a scleroderma mouse model. Elife 2016, 5, e09394.

Round 2

Reviewer 2 Report

The authors have responded to the issues previously raised. I believe the manuscript is improved.